# Turbulent CFD Simulation of Two Rotor-Stator Agitators for High Homogeneity and Liquid Level Stability in Stirred Tank

**DOI:** 10.3390/ma15238563

**Published:** 2022-12-01

**Authors:** Cuicui Yin, Kaihong Zheng, Jiazhen He, Yongnan Xiong, Zhuo Tian, Yingfei Lin, Danfeng Long

**Affiliations:** 1Guangdong Provincial Key Laboratory of Metal Toughening Technology and Application, Institute of New Materials, Guangdong Academy of Sciences, National Engineering Research Center of Powder Metallurgy of Titanium and Rare Metals, Guangzhou 510651, China; 2Institute of Intelligent Manufacturing, Guangdong Academy of Sciences, Guangzhou 510651, China

**Keywords:** solid-liquid mixing, rotor-stator agitator, suspension quality, liquid level stability, power consumption, CFD

## Abstract

Good solid-liquid mixing homogeneity and liquid level stability are necessary conditions for the preparation of high-quality composite materials. In this study, two rotor-stator agitators were utilized, including the cross-structure rotor-stator (CSRS) agitator and the half-cross structure rotor-stator (HCSRS) agitator. The performances of the two types of rotor-stator agitators and the conventional A200 (an axial-flow agitator) and Rushton (a radial-flow agitator) in the solid-liquid mixing operations were compared through CFD modeling, including the homogeneity, power consumption and liquid level stability. The Eulerian–Eulerian multi-fluid model coupling with the RNG *k*–*ε* turbulence model were used to simulate the granular flow and the turbulence effects. When the optimum solid-liquid mixing homogeneity was achieved in both conventional agitators, further increasing stirring speed would worsen the homogeneity significantly, while the two rotor-stator agitators still achieving good mixing homogeneity at the stirring speed of 600 rpm. The CSRS agitator attained the minimum standard deviation of particle concentration *σ* of 0.15, which was 42% smaller than that achieved by the A200 agitators. Moreover, the average liquid level velocity corresponding to the minimum *σ* obtained by the CSRS agitator was 0.31 m/s, which was less than half of those of the other three mixers.

## 1. Introduction

The solid-liquid mixing operation is an important process in the research of particle reinforced metal matrix composites prepared by stirring casting [1]. Due to the obvious density difference between the reinforcing phase and the matrix melt, the problems of particle sinking and agglomeration are serious. In addition, the stability of the liquid surface will also affect the quality of the composite. When the vortex of the liquid surface is large, gas and inclusions will be introduced to pollute the melt. Therefore, good solid-liquid mixing homogeneity and liquid level stability are necessary conditions for the preparation of high-quality composite materials. The design of the agitator needs to meet the following requirements: (1) To provide an intensive melt shearing for the dispersion of agglomerated particles; (2) To generate a relatively homogeneous macro flow in the stirred tank for the homogeneous distribution of particles; (3) To avoid a large liquid level velocity to ensure that gas and other contaminants do not enter the melt from the liquid surface.

The particle suspension is the result of the balance between the driving forces generated by agitator rotation and the particle gravity. In particular, the driving forces of particle suspension are the drag force imposed by the moving fluid and the lifting force generated by the turbulent eddies bursting [2]. The suspension quality of solid particles in the stirred tank is controlled by the process parameters, including particle size/density/loading capacity, melt density/viscosity/liquid level height, stirring speed, size and structure of the agitator, etc. Among them, the optimal design of the agitator determines the flow pattern and turbulence intensity in the stirred tank, which is a significant way to improve the suspension quality of solid particles and to reduce the power consumption [3]. Good uniformity of particle concentration distribution is able to improve product quality and to enhance heat transfer [4,5].

According to the flow patterns, the agitators are mainly classified as radial-flow and axial-flow agitators. Most previous studies have shown that axial-flow agitators are more suitable for solid-liquid mixing than radial-flow agitators [6,7,8]. Among four axial-flow impellers of Lightnin A100, A200, A310 and A320, the A320 was the most effective impeller [9]. The four-bladed 45° PBT impeller was the most energy-efficient comparing with the Lightnin A310 and PF3 impeller [10]. Downward-flow PBT agitators were more efficient than upward-flow PBT agitators [6]. Zhao et al. showed that the blade shape had a great effect on the trailing vortex characteristics and the large curvature led to the longer residence time of the vortex at the impeller tip [11]. In addition to the studies on the performances of agitators with traditional structures, it is believed that many scholars have recently proposed agitators with new structures. The power consumption for the turbine agitator with V cuts has been found to be less than that of the conventional turbine agitator [12]. Mishra et al. demonstrated that the Maxblend impeller attained a higher maximum homogeneity compared to the A200 and the Rushton impellers [13,14]. The punched rigid-flexible impeller was more efficient in suspending solid particles compared with the rigid impeller and rigid-flexible impeller at the same power consumption [15]. The fractal impeller also has reduced power consumption compared to the regular impeller due to the breaking up of the trailing vortices [16,17]. The impeller with zigzag punched blades was developed to enhance the mixing of non-Newtonian fluids by producing impinging jet streams from face-to-face holes [18]. The punched-bionic impellers were proposed to improve the energy efficiency and homogeneity in solid-liquid mixing processes [19].

Rotor-stator agitators are widely used in dispersion, emulsification and homogenization processes because of the high shear rate created by the small rotor-stator gap. The break-up and dispersion of nanoparticle clusters [20,21,22,23], droplet break-up mechanisms [24], droplets size distribution [25], the scale up of the equilibrium drop size [26], the emulsification of a high viscosity oil in water [27] and the dispersion of water into oil [28,29] in rotor–stator mixers were investigated thoroughly by scholars. In contrast, the research on the application of rotor–stator agitators in solid-liquid mixing process is rare. Moreover, most of the current optimum design studies on agitators are aimed at improving the efficiency and homogeneity of solid-liquid mixing, but little attention is paid to the stability of liquid level.

The computational fluid dynamics (CFD) has been widely employed to investigate the solid-liquid mixing process [30,31,32,33,34,35,36,37,38]. There are two main methods for solving the solid-liquid multiphase flow in stirred tank based on Navier-Stokes equations, i.e., Eulerian–Lagrangian (E–L) and Eulerian–Eulerian (E–E) methods [39]. Among them, the E–L method tracks the motion of each particle, which has a large computational resource consumption [40]. The E–E method treats the solid particle mathematically as a continuous phase considering the interpenetration and interaction of the solid and liquid phases [41]. This method has been widely employed by scholars due to the relatively low requirements of computing resources and it was validated by comparing the computational results to the experimental results [10,30,42,43]. Turbulence effects are important for solid-liquid mixing processes and need to be considered in the creating of mathematical models. The Reynolds–averaged Navier–Stokes (RANS) approach has been widely used in large industrial processes due to the good economy and calculation accuracy. The Reynolds stress model establishes six equations for Reynolds stress tensor and one equation for dissipation rate, which can predict all Reynolds stresses correctly, while the equations are difficult to converge. The standard *k–ε* turbulence model estimates the turbulent viscosity by solving the turbulent kinetic energy (k) and the turbulent kinetic energy dissipation rate (ε). However, there exists a large error in the calculation of non-uniform turbulence problems. The RNG *k–ε* turbulence model adopts the framework of two equations, which are derived from the original governing equations of momentum transfer by using the renormalization group method. It is more accurate in predicting the rapidly strained flows by adding an additional term in its ε equation comparing with the standard *k–ε* turbulence model [30]. Siddiqui et al. [44] also demonstrated the RNG *k–ε* model can fairly predict the velocity contours and streamlines. Based on the E–E model along with the RNG *k–ε* turbulence model, scholars have conducted multiple verification work by comparing the simulation data with the experimental results and a large amount of work on multiphase mixing operations ([30,43]).

Most of the current optimum design studies on agitators are aimed at improving the efficiency and homogeneity of solid-liquid mixing, but little attention is paid to the stability of liquid level. In this paper, two types of rotor-stator agitators were utilized to improve the homogeneity and liquid level stability at the same time for the preparation of high-quality composite materials. The flow field, turbulent energy dissipation, pressure field and particle concentration distribution were predicted using the 3D E–E multiphase fluid model along with RNG *k–ε* turbulence model. And the solid-liquid mixing homogeneity, liquid level stability and power consumption of the two types of rotor-stator agitators were evaluated by comparing with those of A200 and Rushton turbine agitators.

## 2. Numerical Modeling

In this investigation, all the numerical simulations are carried out in a stirred tank employing four types of agitators, which including the cross structure rotor-stator (CSRS) agitator, the half cross structure rotor-stator (HCSRS) agitator, the A200 and Rushton turbine, the details of the geometries and dimensions of the stirred tank and agitators are depicted in Figure 1 and Table 1. The specific operating conditions and physical parameters are presented in Table 2. The liquid density and viscosity corresponds to that of AZ91 magnesium alloy at 650 °C, in addition, the particle density corresponds to that of pure titanium.

### 2.1. The Governing Equations

The 3D E–E multiphase model is used for solving the continuity and momentum equations for the liquid and solid phases [45].

The continuity equation is:(1)$$\frac{\partial (\alpha_{q}\rho_{q})}{\partial t}+\nabla \cdot (\alpha_{q}\rho_{q}{\vec{u}}_{q})=0$$
where *α_q_* is the volume fraction of phase *q*, *ρ_q_* is the density, *t* is the time and *u_q_* is the velocity of phase *q*.

The momentum equation for phase *q* is:(2)$$\frac{\partial (\alpha_{q}\rho_{q}{\vec{u}}_{q})}{\partial t}+\nabla \cdot (\alpha_{q}\rho_{q}{\vec{u}}_{q}{\vec{u}}_{q})=-\alpha_{q}\nabla p+\nabla {\bar{\bar{\tau }}}_{q}+\alpha_{q}\rho_{q}\vec{g}+\sum_{p=1}^{n}({\vec{R}}_{pq}+{\dot{m}}_{pq}{\vec{u}}_{pq}-{\dot{m}}_{qp}{\vec{u}}_{qp})$$
where *p* is the pressure, *g* is the gravitational acceleration and *R_pq_* is the interaction force between phase *p* and phase *q*, *m_pq_* represents the mass transfer from phase *p* to phase *q*, *m_qp_* represents the mass transfer from phase *q* to phase *p*, ${\bar{\bar{\tau }}}_{q}$ is the stress-strain tensor defined as:(3)$${\bar{\bar{\tau }}}_{q}=\alpha_{q}\mu_{q}(\nabla {\vec{u}}_{q}+\nabla {\vec{u}}_{q}^{T})+\alpha_{q}(\lambda_{q}-\frac{2}{3}\mu_{q})\nabla {\vec{u}}_{q}Ι$$
here, *μ_q_* and *λ_q_* are the shear and bulk viscosity of phase *q*, respectively.

The interaction force between phase *p* and phase *q* is computed as follows:(4)$$\sum_{p=1}^{n}{\vec{R}}_{pq}=\sum_{p=1}^{n}k_{pq}({\vec{u}}_{p}-{\vec{u}}_{q})$$
where *k_pq_* is the momentum exchange coefficient between phase *p* and phase *q*. The fluid-solid momentum exchange is mainly controlled by drag, lift, virtual mass and Basset forces. Studies have shown that the lift, virtual mass and Basset forces are irrespective for modeling the solid holdup profiles [46]. Hence, only the drag force is considered in the current calculation. The *k_pq_* is obtained by the Gidaspow model [41]:(5)$$k_{pq}=\left\{\begin{array}{l} \frac{3}{4}C_{D}\frac{\alpha_{p}\alpha_{q}\rho_{p}\left|{\vec{u}}_{p}-{\vec{u}}_{q}\right|}{d_{p}}\alpha_{q}^{-2.65}\begin{array}{cc} & \left(\alpha_{q}>0.8\right) \end{array}\\ \frac{150\alpha_{p}(1-\alpha_{q})\mu_{q}}{\alpha_{q}d_{p}^{2}}+\frac{1.75\alpha_{p}\rho_{q}\left|{\vec{u}}_{p}-{\vec{u}}_{q}\right|}{d_{p}}\begin{array}{cc} & \left(\alpha_{q}\leq 0.8\right) \end{array} \end{array}\right.\;$$
where *d_p_* is the particle diameter and *C_D_* is the drag coefficient, which can be written as:(6)$$C_{D}=\frac{24\left[1+0.15(\alpha_{q}Re{)}^{0.687}\right]}{\alpha_{q}Re}$$
where *Re* is the relative Reynolds number which can be computed via:(7)$$Re=\frac{\rho_{q}d_{p}\left|{\vec{u}}_{p}-{\vec{u}}_{q}\right|}{\mu_{q}}$$

The RNG *k*–*ε* turbulence model [47] is employed for simulating the turbulence effect. The turbulent kinetic energy (*k*) and the turbulent kinetic energy dissipation rate (*ε*) are expressed as follows:(8)$$\frac{\partial }{\partial t}(\rho k)\;+\;\frac{\partial }{\partial x_{i}}(\rho ku_{i})\;=\;\frac{\partial }{\partial x_{j}}(\alpha_{k}\mu_{t}\frac{\partial k}{\partial x_{j}})-\rho \bar{{u^{\prime }}_{i}{u^{\prime }}_{j}}\frac{\partial u_{j}}{\partial x_{i}}-\rho \varepsilon -2\rho \varepsilon \frac{k}{a^{2}};$$
(9)$$\frac{\partial }{\partial t}(\rho \varepsilon )\;+\;\frac{\partial }{\partial x_{i}}(\rho \varepsilon u_{i})\;=\;\frac{\partial }{\partial x_{j}}(\alpha_{\varepsilon }\mu_{t}\frac{\partial \varepsilon }{\partial x_{j}})-C_{1\varepsilon }\frac{\varepsilon }{k}\rho \bar{{u^{\prime }}_{i}{u^{\prime }}_{j}}\frac{\partial u_{j}}{\partial x_{i}}-C_{2\varepsilon }\rho \frac{\varepsilon^{2}}{k}-R_{\varepsilon }.$$
where *μ_t_* is the turbulent viscosity and *R_ε_* is the additional term in the *ε* equation that takes:(10)$$\mu_{t}\;=\;\rho C_{\mu }\frac{k^{2}}{\varepsilon }f(\alpha_{s},\Omega ,\frac{k}{\varepsilon });$$
(11)$$R_{\varepsilon }\;=\;\frac{C_{\mu }\rho \eta^{3}(1-\eta /\eta_{0})}{1+\beta \eta^{3}}\frac{\varepsilon^{2}}{k}.$$
where *C_μ_* is 0.0845 derived by RNG theory, *α_s_* is a swirl constant depending on the strength of swirling flows and is set to 0.07 for mildly swirling flows, Ω is a characteristic swirl number evaluated within Fluent, *β* = 0.012, *η*_0_ = 4.38, *η* = *S_k_*/*ε*, *S* is the total entropy. The constants *C*_1*ε*_ = 1.42 and *C*_2*ε*_ = 1.68 are used by default.

### 2.2. Simulation Setup

The simulations of solid-liquid mixing in the stirred tank are conducted using the CFD method. The 3D Eulerian–Eulerian multiphase model along with the RNG *k–ε* turbulence model are used for simulating the granular flow and the turbulence effects. Moreover, the moving reference frame (MRF) technique is employed in this work [48]. The computational domain can be divided into stationary and moving zones, and the moving zone is specified frame motion at a rotating speed. The near wall zones are modeled by employing the enhanced wall function. The unstructured tetrahedral grids are used throughout the computational domain, and the grid distribution in the stirred tank is shown in Figure 2. The grid independence test is carried out by using three sets of grids: 107,732 elements, 263,709 elements, 439,085 elements, 599,290 elements, 827,636 elements and 1,179,019 elements, as shown in Figure 3. The deviation of the relative standard deviation of particle concentration *σ* and the mean velocity in the computational domain between 599,290 and 1,179,019 cells are less than 1%. Thus, the grid of 599,290 elements is used in the current simulations. The solid particles were initially patched in a defined region at the bottom of tank. The convergence criterion of all transport equations is specified as the residual values below 0.0001. The calculated time step is set to 0.001 s and the maximum number of iterations to be performed per time step is 20. A typical simulation with such grid resolution takes about 72 h real time with paralleling 32 Intel Xeon CPU cores (2.3 GHz).

To evaluate the homogeneity of solid particles distribution in the stirred tank, the suspension quality is defined by the relative standard deviation of particle concentration *σ* that can be expressed as [49]:(12)$$\sigma =\sqrt{\frac{1}{n}\sum_{i=1}^{n}{\left(\frac{\alpha_{i}-\alpha_{avg}}{\alpha_{avg}}\right)}^{2}}$$
where *α_avg_* is the averaged volume fraction of solid particles in the whole computational domain, *i* denotes the discrete volume element, n is the total number of computational cells. Homogeneous suspension conditions can be achieved when *σ* < 0.2, while the incomplete suspension is resulted at *σ* > 0.8. The mixing energy level (*MEL*) denotes the power consumption per unit volume, which is calculated as:(13)MEL=P/v
where *v* is the volume of working fluids, and *P* is the power consumption of agitator which can be calculated by:(14)P=2πNT/60
where *N* is the stirring speed, *T* is the torque of agitator.

### 2.3. Model Validation

To validate the accuracy of the current numerical model, we simulate the dispersions and holdups of sand in the glycerite–sand system from the experiments of Wang et al. [30]. Figure 4 shows the comparison of numerical and experimental results for axial profiles of the normalized solid volume fraction. The numerical results have showed a good agreement with the experimental data. The slight deviations could root in the experimental error of solid holdup which is basically generated from the fluctuation of the turbulent flow and the sampling [30]. Therefore, the present liquid-solid multiphase model can predict well the distribution of particles in the stirred tank.

## 3. Results and Discussion

### 3.1. Flow Fields

First, the flow fields of the two new rotor-stator agitators of CSRS and HCSRS with that of the conventional agitators of the A200 and Rushton are compared. Figure 5 shows the velocity vectors of solid particles for the four types of agitators, the color and arrows in the figures indicate the magnitude and direction of the velocity vector, respectively. For the CSRS agitator, as shown in Figure 5a, after the fluid is discharged through the blade tip, part of the fluid forms a squeeze flow in the gap between the stator and rotor and circulates inside the stator. Part of the fluid passes through the circular hole of the stator structure after being discharged by the paddle (see Figure 6a), then impacts the side wall of the stirred tank and is divided into two upward and downward streams. The upper stream forms a small annular flow and merges with the lower fluid and finally enters the stator under the agitator to form a complete circulation. It can be seen that the velocity distribution in the whole stirred tank is relatively homogeneous, and there is almost no flow dead zone. For the HCSRS agitator, as described in Figure 5b, the fluid is discharged obliquely downward after being discharged from the blade tip. After being blocked by the wall surface, it forms a large circulation upward along the side wall of the stirred tank, and then enters the stator through the circular holes on the side wall of the stator structure (see Figure 6b). A small part of the fluid forms an internal circulation above the rotor, and most of the fluid flows to the rotor, forming a complete flow cycle. The velocity distribution in the whole stirred tank is relatively homogeneous, but the flow at the bottom of the agitator is weak. For the A200 agitator, as exhibited in Figure 5c, after the fluid is discharged from the tip of the paddle, it will obliquely impact the side wall of the stirred tank and flow upward along the side wall to form a circulation. It can be seen that the flow under the agitator, near the stirring rod and above the stirred tank is very weak, and there are small vortices near the wall surface above the stirred tank, which are not conducive to solid-liquid mixing. For the Rushton agitator, as descripted in Figure 5d, the fluid is discharged from the tip of the blade and impinges on the side wall of the stirred tank. It is divided into two streams, one of which forms a large circulation above the agitator, and the other forms a small circulation below the agitator along the wall surface. Similar to the A200 agitator, the flow under the agitator, near the agitator bar and above the agitator tank is very weak, and there are small vortices on the wall above the agitator tank. In addition, the low-speed zone near the agitator shaft is wider, which is not conducive to solid-liquid mixing. In conclusion, the HCSR agitator has a more homogeneous velocity distribution, which is conducive to the homogeneous dispersion of particles in the melt.

The stability of liquid surface is very important to the quality of composite materials in the process of stirring preparation. Stable liquid surface can ensure that gas and other contaminants cannot enter the melt from the liquid surface. Thus, the liquid level velocity for the four types of agitators are investigated in this study, as shown in Figure 7. It can be attained that the liquid level velocities obtained by the two new types of rotor-stator agitators are far lower than that obtained by the two conventional agitators. Additionally, the liquid level velocity obtained by the CSRS agitator is the minimum, while the liquid level velocity obtained by the Rushton agitator is the maximum. Since the pressure distributions outside the stator of the rotor-stator agitators are more uniform, and the specific pressure distribution will be discussed later.

### 3.2. Turbulent Energy Dissipation

Turbulent energy dissipation rate is an important parameter in particle dispersion system, which determines the size distribution of bubbles, droplets and agglomerated particles. Therefore, the turbulent energy dissipation distributions on the vertival center plane and horizontal plane across the rotor at the stirring speed of 400 rpm for the four types of agitators were explored, as shown in Figure 8 and Figure 9. It can be found that, for the two new types of rotor-stator agitators, the high turbulent energy dissipation areas are mainly distributed in the gaps between the stator and rotor and the circular hole channels on the stator. Moreover, the turbulent energy dissipation near the inner wall of the solid connection part between the circular holes on the stator is the highest due to the strong shear effect on the fluid between the high-speed rotating rotor and the stationary wall. For the CSRS agitator, the turbulent energy dissipation in the area between the upper surface of the rotor blade and the inner wall at the top of the stator is also very high. For the two conventional agitators, the high turbulent energy dissipation area is located at the place where the blade tips pass through, and the turbulent energy dissipation generated by A200 agitator is significantly higher than that generated by Rushton agitator. In a comprehensive comparison, the high turbulent energy dissipation areas generated by the two new types of rotor-stator agitators are significantly more than those generated by the two conventional agitators and the high turbulent energy dissipation areas generated by the CSRS agitator distribute more widely.

### 3.3. Pressure Fields

In order to better understand the reasons why different agitators produce different flow patterns, the pressure fields of the four agitators are investigated, as shown in Figure 10. For the CSRS agitator (see Figure 10a), the fluid follows the rotation of the rotor and passes through the circular holes on the side wall of the stator under the action of centrifugal force. Since the external fluid cannot enter the stator structure in time, a large negative pressure is formed inside the stator structure. The fluid outside the stator enters the stator from the opening below the stator under the pressure difference, which is also the reason for the large upward speed near the middle and lower parts of the rotor. For the HCSRS agitator (see Figure 10b), the fluid flows out from the opening below the stator as the blade rotates. Since the external fluid cannot enter the stator structure in time, an obvious negative pressure field appears in the area above the rotor near the mixing shaft. Therefore, the fluid around the stator enters the stator through the circular holes on the stator side wall under the pressure difference. For the A200 agitator and the Rushton agitator, as exemplified in Figure 10c,d, the fluid is pushed to the side wall of the stirred tank under the action of the rotor. The pressure distribution in the stirred tank is gradually increased from the center of the stirred tank to the outside, which is detrimental to the liquid level stability. The Rushton agitator has a stronger radial liquid discharge capacity, resulting in greater pressure in the stirred tank. For CSRS agitator, there is an obvious pressure gradient inside the stator, but the pressure distribution outside the stator is very uniform, which is very conducive to obtaining a stable liquid level.

### 3.4. Solid Particle Distribution

To compare the performances for the four types of agitators, the distribution of solid volume fraction at the vertical and horizontal planes are exhibited in Figure 11 and Figure 12. Additionally, the quantitative study on the axial distribution of solid volume fraction is shown in Figure 8. The averaged solid volume fraction of the horizontal plane (*α_ave-p_*) is normalized by that of the whole stirred tank (*α_ave_*). For the CSRS agitator, it can be seen that the solid particles are evenly distributed throughout the stirred tank due to the relatively homogeneous velocity distribution (as shown in Figure 11a), but the concentration of particles increases slightly at the bottom of the stirred tank and the problem of particle accumulation is more serious in the bottom central area (as shown in Figure 12a). For the HCSRS agitator, the distribution of solid particles in the whole stirred tank is also basically homogeneous (as shown in Figure 11b), except that the particles in the bottom central area of the stirred tank are relatively sparse and the concentration of particles increases slightly above the stirred tank. The closer the particles are to the wall, the higher the concentration can be seen from the radial distribution (as shown in Figure 12b). For the A200 agitator, the distribution of particles in the stirred tank is not homogeneous, the particles are heavily deposited at the bottom of the stirred tank due to the flow dead zone, and the particles near the stirring shaft are extremely sparse due to the weak flow (as shown in Figure 11c and Figure 12c). For the Rushton agitator, the distribution of particles in the stirred tank is very uneven, the particles accumulate slightly under the impeller blades, and the particles near the stirring shaft and below the stirring shaft are extremely sparse (as shown in Figure 11d). It can be seen from the radial distribution that particles can also accumulate near the wall above the stirred tank (as shown in Figure 12d). In conclusion, the use of two rotor-stator agitators can significantly improve the homogeneity of solid particle distribution in the stirred tank compared with the conventional agitators of A200 and Rushton agitators. Similar conclusions can also be obtained by quantitative analysis of the axial distribution of solid particles, as exhibited in Figure 13.

### 3.5. Effect of Stirring Speed

It is known that stirring speed has a great influence on the homogeneity of solid-liquid mixing. Figure 14 compares the relative standard deviation of particle concentration of four agitators at different stirring speeds. It can be found that the minimum *σ* achieved by the two rotor-stator agitators of CSRS and HCSRS are much lower than that of the two conventional agitators of A200 and Rushton. The CSRS agitator yielded the best solid-liquid mixing homogeneity, while the Rushton agitator yielded the worst. For all agitators, the influence of stirring speed on the homogeneity of solid-liquid mixing shows a rule of first promoting and then suppressing. However, for the two conventional agitators, after achieving the best mixing homogeneity, increasing the stirring speed will worsen the mixing homogeneity significantly. For the two rotor-stator agitators, after achieving the best mixing homogeneity, increasing the stirring speed has little effect on the mixing homogeneity, especially for the CSRS agitator. It is very important for the dispersion of agglomerated particles and the grain refinement to enhance the shearing effect by increasing the stirring speed, while ensuring a good homogeneity of solid-liquid mixing in the stirring casting process.

### 3.6. Evaluation of Power Consumption and Surface Stability

In order to evaluate the performances of the two rotor-stator agitators in solid-liquid mixing operation, the solid-liquid mixing homogeneity verse the power consumption for the four agitators was investigated, as shown in Figure 15. It can be found that the power consumption required by the conventional agitators to achieve the minimum *σ* are much lower than that of the two rotor-stator agitators, because the addition of the stator structure increases the flow resistance, which in turn significantly increases the power consumption required to achieve the same speed. At low power consumption, the two new agitators could not perform very well and the resulting homogeneity of solid-liquid mixing is inferior to that of the two conventional agitators. When the optimum solid-liquid mixing homogeneity is achieved in both conventional agitators, increasing power consumption will worsen the homogeneity significantly, while the two new agitators will continue to optimize the mixing homogeneity, even when the optimum mixing homogeneity is reached, the influence of increasing power consumption on the mixing homogeneity is very small. In addition, at the same power consumption, the homogeneity of solid-liquid mixing obtained by A200 is always lower than that of Rushton.

The liquid level velocities corresponding to the four types of agitators when reaching the minimum standard deviation of particle concentration are evaluated in Figure 16. The two rotor-stator agitators can not only obtain better mixing homogeneity due to the more homogeneous velocity distributions in the whole stirring tanks, but also gain lower liquid level velocities due to the more uniform pressure distributions outside the stators. Among them, the liquid level stability of the CSRS agitator is much better than that of the other three agitators.

## 4. Conclusions

The performances of the two rotor-stator agitators (CSRS and HCSRS) for solid-liquid mixing operations in the stirred tank comparing with that of the conventional agitators of A200 and Rushton were investigated using the CFD modeling to improve the mixing homogeneity and liquid level stability.

The minimum *σ* achieved by the two rotor-stator agitators were much lower than that achieved by the two conventional agitators. The CSRS agitator yielded the best solid-liquid mixing homogeneity due to the relatively homogeneous velocity distribution in the whole stirred tank. For all agitators, the influence of stirring speed on the homogeneity of solid-liquid mixing showed a rule of first promoting and then suppressing. When the optimum solid-liquid mixing homogeneity was achieved in both conventional agitators, increasing stirring speed would worsen the homogeneity significantly, while the two rotor-stator agitators still achieving good mixing homogeneity under high shear conditions. In addition, the high turbulent energy dissipation areas generated by the CSRS agitator were the most widely distributed, which was very advantageous to particle dispersion and grain refinement during the preparation of composite materials.

The liquid level velocities corresponding to the minimum standard deviation of particle concentration obtained by the two rotor-stator agitators were also far less than that obtained by the two conventional agitators. The specific numerical order is: CSRS < HCSRS < A200 < Rushton. The above findings are of great significance for promoting the solid-liquid mixing homogeneity, enhancing the shear effect on the agglomerated particles and melt and stabilizing the liquid surface, thereby improving the quality of the composite materials.

## Figures and Tables

**Figure 1 materials-15-08563-f001:**
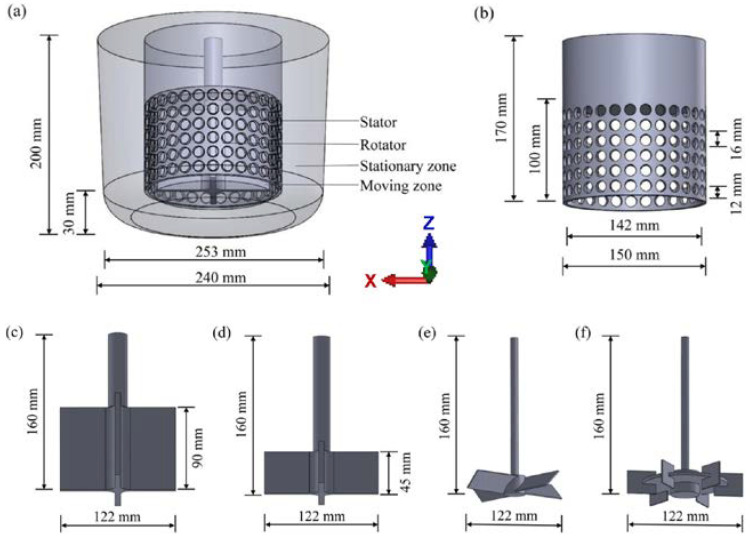
Geometries and dimensions of the stirred tank and agitators; (**a**) computational domain of the stirred tank; (**b**) the stator of the rotor-stator agitators; (**c**) the rotor of the CSRS agitator; (**d**) the rotor of the HCSRS agitator; (**e**) A200 agitator; (**f**) Rushton turbine agitator.

**Figure 2 materials-15-08563-f002:**
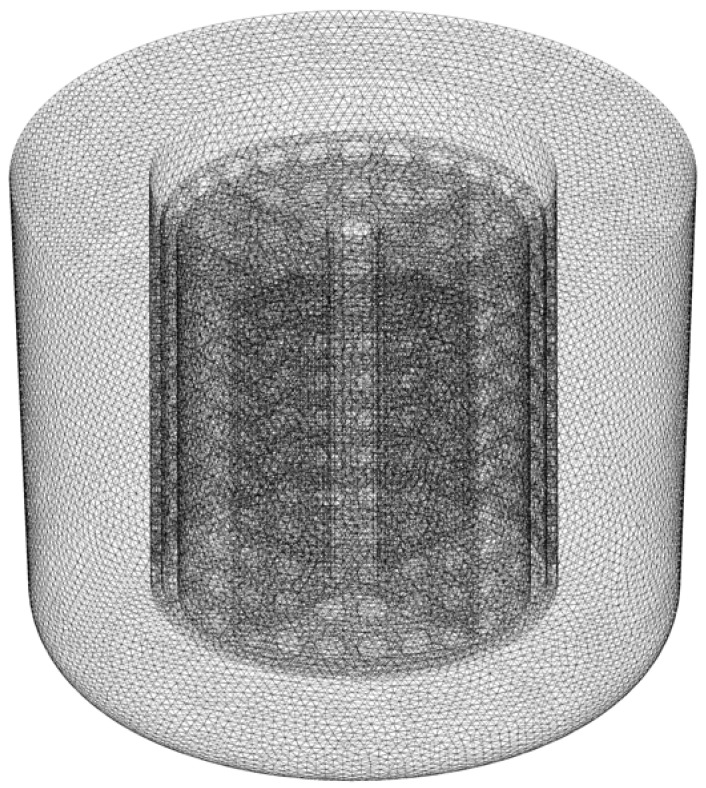
Grid distribution in the stirred tank.

**Figure 3 materials-15-08563-f003:**
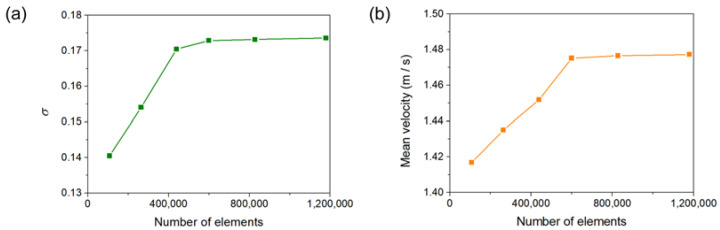
Grid independence test; (**a**) Effect of the number of elements on the relative standard deviation of particle concentration; (**b**) Effect of the number of elements on the mean velocity of the computational domain.

**Figure 4 materials-15-08563-f004:**
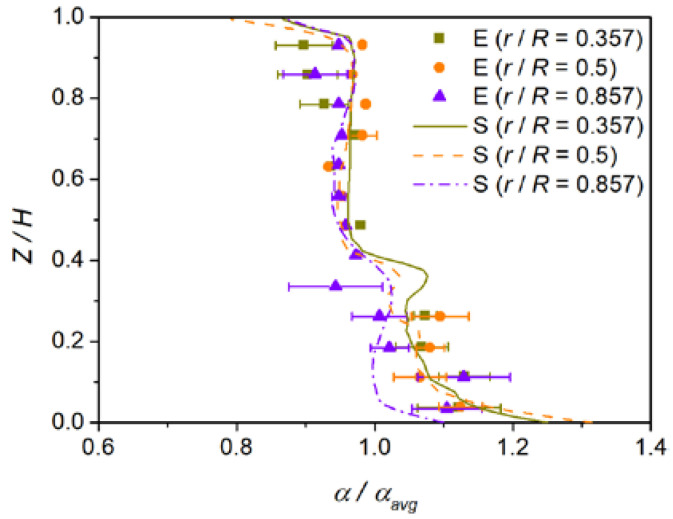
Comparison of numerical and experimental [30] results for axial profiles of the normalized solid volume fraction.

**Figure 5 materials-15-08563-f005:**
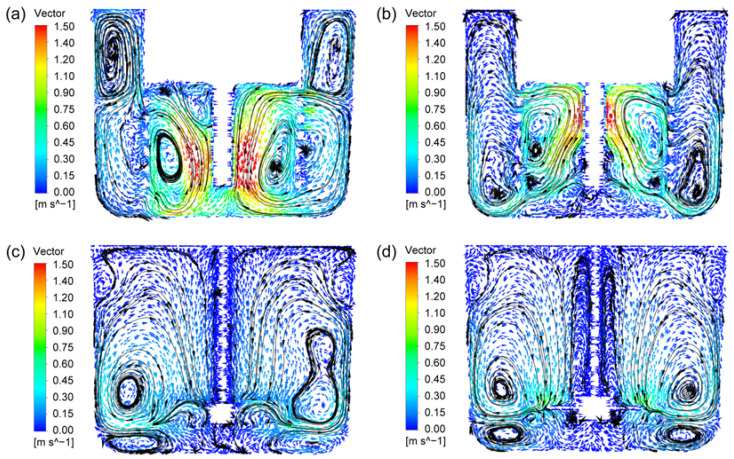
Velocity vectors of solid particles for the four types of agitators at the stirring speed of 400 rpm; (**a**) CSRS agitator; (**b**) HCSRS agitator; (**c**) A200 agitator; (**d**) Rushton agitator.

**Figure 6 materials-15-08563-f006:**
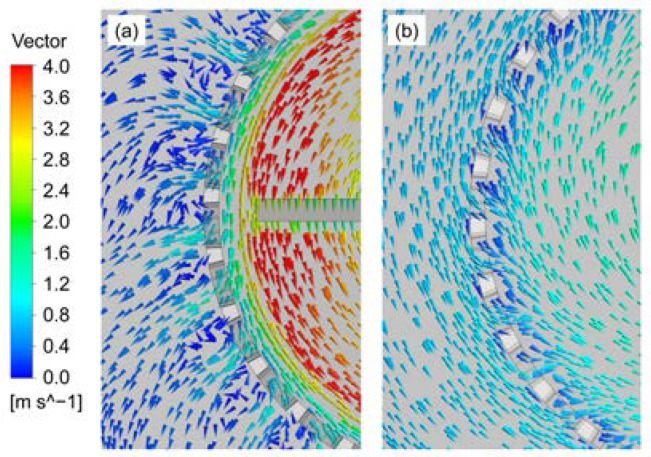
Velocity vectors near the stator sidewall holes for the two types of rotor-stator agitators at the horizontal plane of *Z* = 0.105 m under the stirring speed of 400 rpm; (**a**) CSRS agitator; (**b**) HCSRS agitator.

**Figure 7 materials-15-08563-f007:**
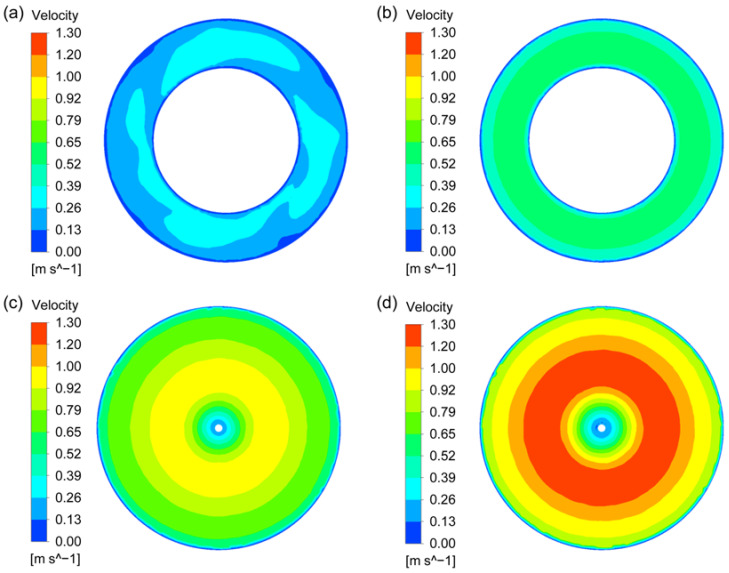
Liquid level velocity for the four types of agitators at the stirring speed of 400 rpm; (**a**) CSRS agitator; (**b**) HCSRS agitator; (**c**) A200 agitator; (**d**) Rushton agitator.

**Figure 8 materials-15-08563-f008:**
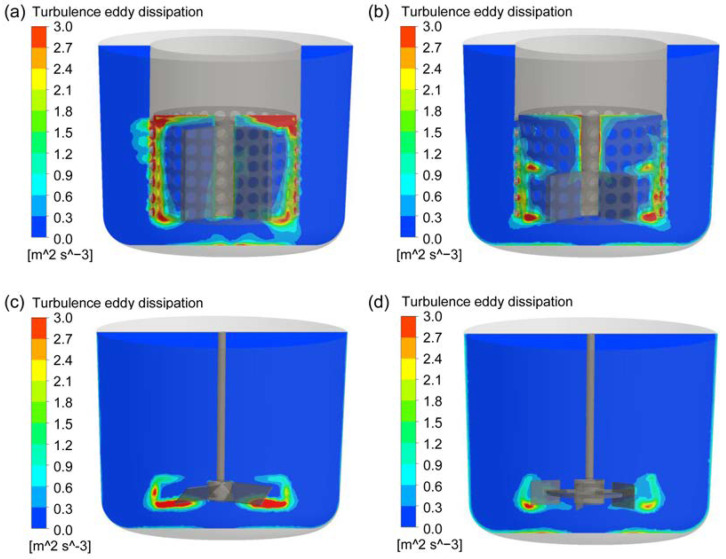
The distribution of turbulent energy dissipation rate on the vertival plane at the stirring speed of 400 rpm for the four types of agitators; (**a**) CSRS agitator; (**b**) HCSRS agitator; (**c**) A200 agitator; (**d**) Rushton agitator.

**Figure 9 materials-15-08563-f009:**
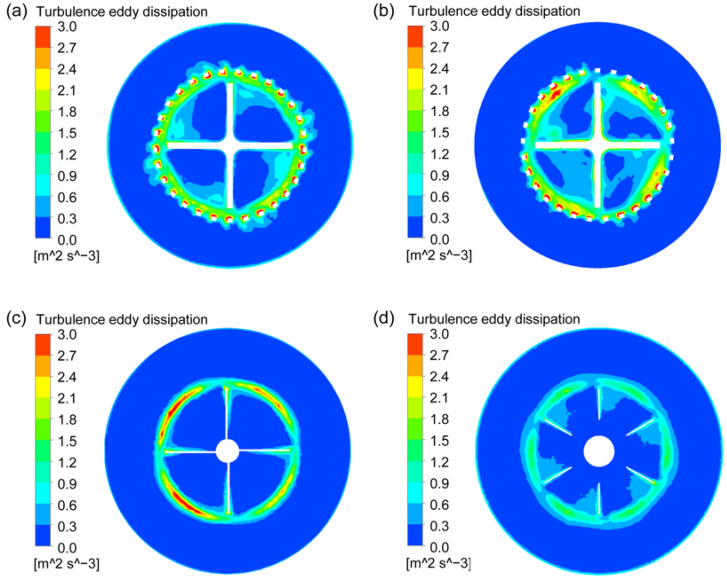
The distribution of turbulence energy dissipation rate on the horizontal plane (*Z* = 0.038 m) at the stirring speed of 400 rpm for the four types of agitators; (**a**) CSRS agitator; (**b**) HCSRS agitator; (**c**) A200 agitator; (**d**) Rushton agitator.

**Figure 10 materials-15-08563-f010:**
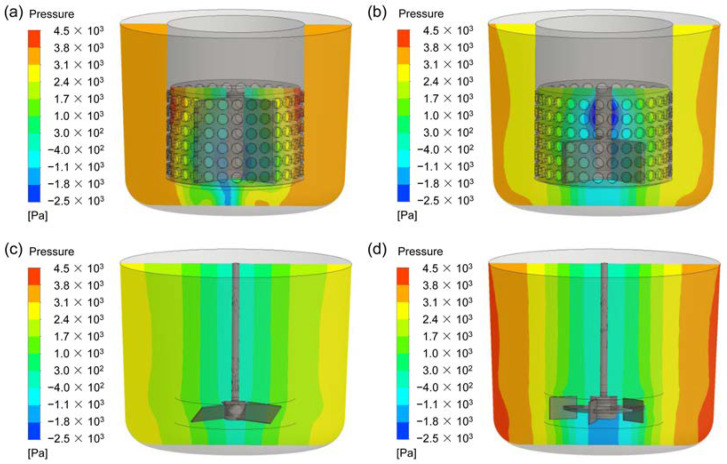
Pressure fields at the vertical plane for the four types of agitators at the stirring speed of 400 rpm; (**a**) CSRS agitator; (**b**) HCSRS agitator; (**c**) A200 agitator; (**d**) Rushton agitator.

**Figure 11 materials-15-08563-f011:**
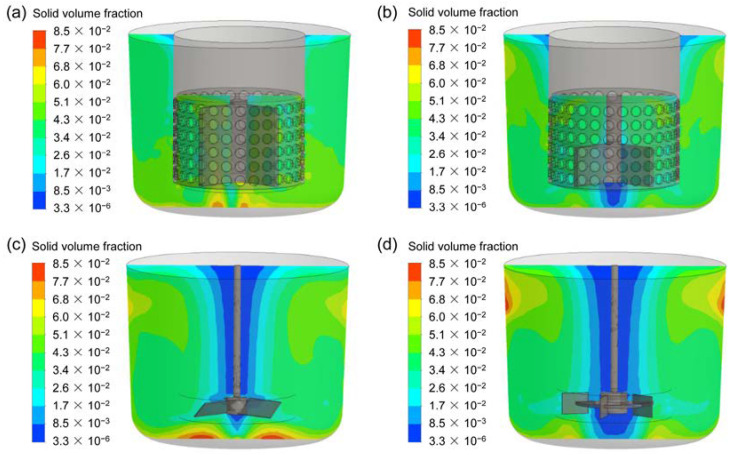
The distribution of solid volume fraction at the vertical plane for the four types of agitators at the stirring speed of 400 rpm; (**a**) CSRS agitator; (**b**) HCSRS agitator; (**c**) A200 agitator; (**d**) Rushton agitator.

**Figure 12 materials-15-08563-f012:**
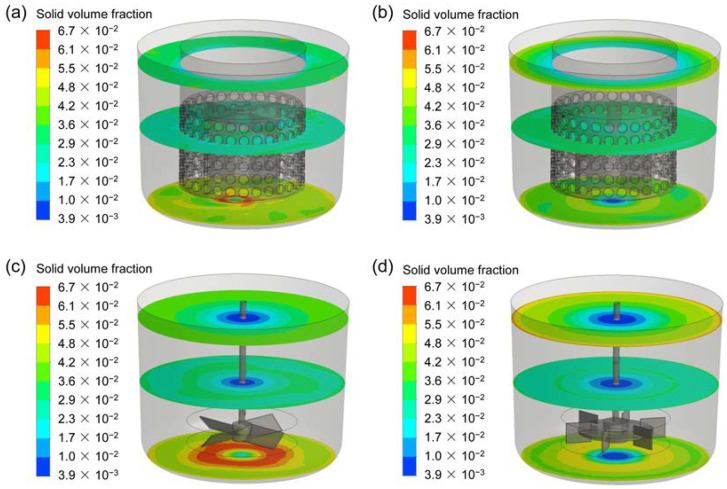
The distribution of solid volume fraction at different horizontal planes for the four types of agitators at the stirring speed of 400 rpm; (**a**) CSRS agitator; (**b**) HCSRS agitator; (**c**) A200 agitator; (**d**) Rushton agitator.

**Figure 13 materials-15-08563-f013:**
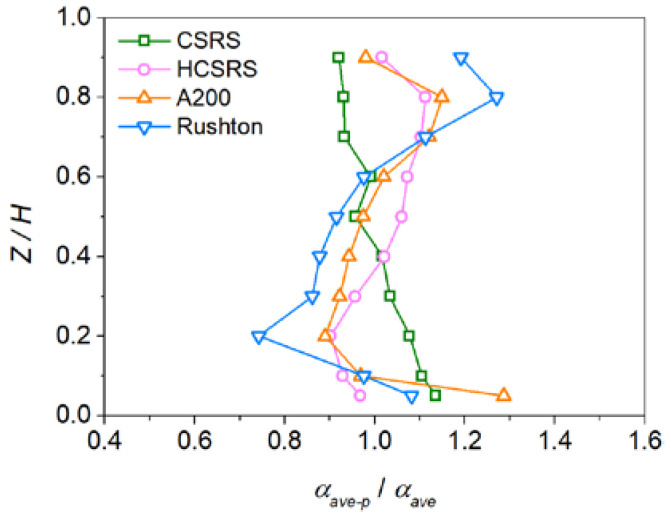
The axial distribution of the normalized averaged solid volume fraction on the horizontal planes (*α_ave-p_*) for the four types of agitators at the stirring speed of 400 rpm.

**Figure 14 materials-15-08563-f014:**
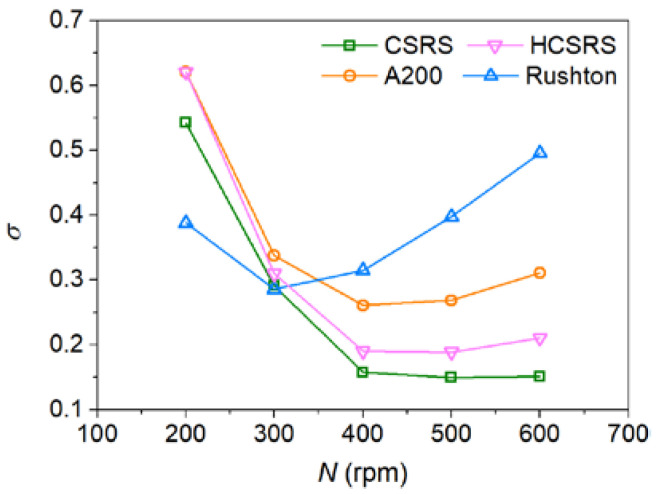
Effect of stirring speed on the relative standard deviation of particle concentration for the four types of agitators.

**Figure 15 materials-15-08563-f015:**
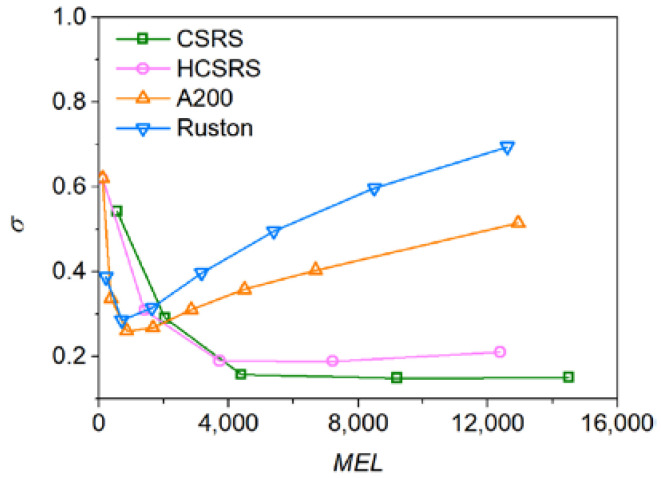
Effect of power consumption per unit volume on the relative standard deviation of particle concentration for the four types of agitators.

**Figure 16 materials-15-08563-f016:**
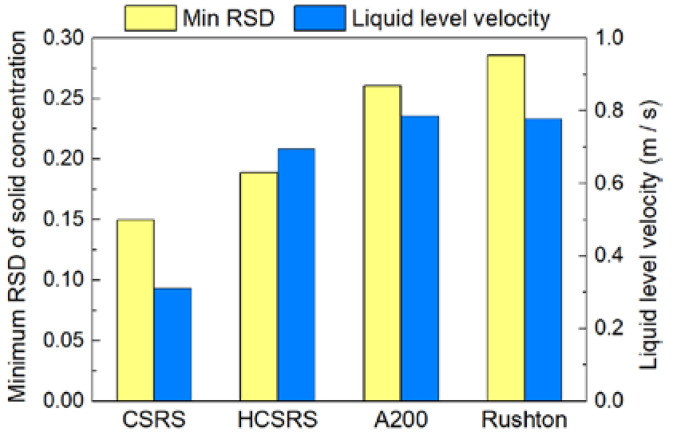
The minimum relative standard deviation of particle concentration and the corresponding average liquid level velocity obtained by the four agitators.

**Table 1 materials-15-08563-t001:** Geometric parameters of the stirred tank and agitators.

Geometric Parameter (mm)	Values
Diameter of top of stirring tank	253
Diameter below stirring tank	240
Stirring tank height	200
Height of hole area	100
Total height of stator	170
Hole center spacing of stator	16
Hole diameter of stator	12
Inner diameter of stator	142
Outer diameter of stator	150

**Table 2 materials-15-08563-t002:** Operating conditions and physical parameters.

Variable	Values
Agitator type	CSRS, HCSRS, A200, Rushton
Stirring speed (rpm)	200, 300, 400, 500, 600
Particle size (μm)	100
Solids weight fraction (wt.%)	10
Grain density (kg/m^3^)	4506
Liquid density (kg/m^3^)	1650
Liquid viscosity (Pa*s)	0.00139

## Data Availability

Not applicable.

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
