# Peer review of "Turbulent CFD Simulation of Two Rotor-Stator Agitators for High Homogeneity and Liquid Level Stability in Stirred Tank"

_materials, 2022, doi:10.3390/ma15238563_

Round 1

Reviewer 1 Report

The following comments have been made to improve the quality of the manuscript.

1)    The last four lines in the abstract are redundant. You should write the exact outcome of the work here. Mention the main outcomes in terms of numbers.

2)    While reading your literature review in the introduction section, I barely see any discussion on the research gap. Discuss previous work in detail and mention the research gaps.

3)    Did you mention the choice of turbulence model?

4)    How you have decided best turbulence model? The following research work may be helpful to you: https://doi.org/10.2355/isijinternational.54.2578

5)    Lines 79 to 88 should be written in a better way. You need to present your objectives and methodology here.

6)    Mention all CFD governing equations in section 2.

7)    Mention the equations for the turbulence model and other appropriate equations. See the following work:  https://doi.org/10.3390/ma15217458

8)    Please mention the geometrical aspect before you start a discussion on mathematical equations.

9)    Did you study the grid test before the simulation? Can you explain how grid elements can affect turbulence modeling results?

10)        The validation in Figure 1 is not clear. Make legends in such a way that we can see the difference between experimental and numerical work.

11)        Check line 189 (“Our project is to prepare titanium particle reinforced magnesium matrix composites 189 by stirring casting method.”)

12)        There are many grammatical mistakes in this work.

Author Response

Point 1: The last four lines in the abstract are redundant. You should write the exact outcome of the work here. Mention the main outcomes in terms of numbers.

Response 1: Following the reviewer’s suggestion, the abstract has been revised, as described in the Line 10-24 in the revised manuscript.

Point 2: While reading your literature review in the introduction section, I barely see any discussion on the research gap. Discuss previous work in detail and mention the research gaps.

Response 2: Following the reviewer’s suggestion, the research gaps have been added, as described in the Line 95-96 and Line 126-128 in the revised manuscript.

Point 3: Did you mention the choice of turbulence model?

Response 3: Following the reviewer’s suggestion, the choice of turbulence model have been added, as described in the Line 2-3 in the revised manuscript.

Point 4: How you have decided best turbulence model? The following research work may be helpful to you: https://doi.org/10.2355/isijinternational.54.2578

Response 4: Following the reviewer’s suggestion, the recommended references have been added, which is helpful to the choice of turbulence model, as described in the Line 108-125 in the revised manuscript.

Point 5: Lines 79 to 88 should be written in a better way. You need to present your objectives and methodology here.

Response 5: Following the reviewer’s suggestion, this section has been revised and my objectives and methodology have been added, as described in the Line 126-134 in the revised manuscript.

Point 6: Mention all CFD governing equations in section 2.

Response 6: Following the reviewer’s suggestion, all CFD governing equations used in my study have been added, as described in the Line 161-188 in the revised manuscript.

Point 7: Mention the equations for the turbulence model and other appropriate equations. See the following work:  https://doi.org/10.3390/ma15217458

Response 7: Following the reviewer’s suggestion, equations of the turbulence model have been added, as described in the Line 181-188 in the revised manuscript.

Point 8: Please mention the geometrical aspect before you start a discussion on mathematical equations.

Response 8: Following the reviewer’s suggestion, the geometrical aspect has been mentioned  before the discussion on mathematical equations, as described in the Line 146-160 in the revised manuscript.

Point 9: Did you study the grid test before the simulation? Can you explain how grid elements can affect turbulence modeling results?

Response 9: The grid test was studied before the simulation, and the effect of grid elements on the mean velocity has been added, as described in the Line 226-229 in the revised manuscript.

Point 10: The validation in Figure 1 is not clear. Make legends in such a way that we can see the difference between experimental and numerical work.

Response 10: Following the reviewer’s suggestion, the range of coordinate in the figure has been revised, as described in the Line 2-3 in the revised manuscript.

Point 11: Check line 189 (“Our project is to prepare titanium particle reinforced magnesium matrix composites by stirring casting method.

Response 11: Following the reviewer’s suggestion, this sentence has been revised, as described in the Line 297-299 in the revised manuscript.

Point 12: There are many grammatical mistakes in this work.

Response 12: Following the reviewer’s suggestion, the grammar of the entire content has been checked and revised.

Reviewer 2 Report

Authors performed a computational simulation of two rotor-stator agitators for high homogeneity and liquid level stability in stirred tanks. The study needs following revisions by using Eulerian-Eulerian multi-fluid model coupling with the RNG k-ε 17 turbulence model. 

The title can be revised as 

Turbulent CFD simulation of two rotor-stator agitators for high homoge- 2 neity and liquid level stability in stirred tanks

- Grid distribution and wide grid tests are necessary in the study.

-coordinates must be given in Fig. 2.

-Some values need references in Table 2.

- Introduction paragraph of conclusion is very long.

-Physical comments of findings can be extended. 

- Flow field by lines can be added. 

Introduction can be extended by adding some related works. For example,

https://doi.org/10.1016/j.ijheatmasstransfer.2014.07.031

- Please check the explanations for all abbreviations. 

Then, manuscript can be accepted. 

Author Response

Point 1: The title can be revised as “Turbulent CFD simulation of two rotor-stator agitators for high homogeneity and liquid level stability in stirred tanks”.

 Response 1: Following the reviewer’s suggestion, the title has been revised as “Turbulent CFD simulation of two rotor-stator agitators for high homogeneity and liquid level stability in stirred tanks”, as described in the Line 2-3 in the revised manuscript.

Point 2: Grid distribution and wide grid tests are necessary in the study.

Response 2: Following the reviewer’s suggestion, the grid distribution has been added, as described in the Line 244-245 in the revised manuscript. And the wide grid tests have been done, as described in the Line 226-229 in the revised manuscript.

Point 3: Coordinates must be given in Fig. 2.

 Response 3: Following the reviewer’s suggestion, the coordinate has been added in the figure, as described in the Line 155 in the revised manuscript.

Point 4: Some values need references in Table 2.

Response 4: Following the reviewer’s suggestion, some values of geometric parameters have been referenced in Table 1, as described in the Line 159 in the revised manuscript.

Point 5: Introduction paragraph of conclusion is very long.

 Response 5: Following the reviewer’s suggestion, the introduction paragraph of conclusion has been refined, as described in the Line 452-455 in the revised manuscript.

Point 6: Physical comments of findings can be extended.

Response 6: Following the reviewer’s suggestion, some physical comments of findings have been extended, as described in the Line 321-323, Line 356-360, Line 370-371, Line 380-381 and Line 443-446 in the revised manuscript.

Point 7: Flow field by lines can be added.

 Response 7: Following the reviewer’s suggestion, the flow fields by lines have been add in Figure 5, as described in the Line 290 in the revised manuscript.

Point 8: Introduction can be extended by adding some related works. For example, https://doi.org/10.1016/j.ijheatmasstransfer.2014.07.031

Response 8: Following the reviewer’s suggestion, the introduction has been extended by adding some related works, as described in the Line 65-66 in the revised manuscript.

Point 9: Please check the explanations for all abbreviations.

Response 9: Following the reviewer’s suggestion, the explanations of all abbreviations have been checked and revised, as described in the Line 12-13 and Line 149 in the revised manuscript.

Round 2

Reviewer 1 Report

Received.